

# Overview: 'Global change effects on terrestrial biogeochemistry at the plant-soil interface'

Lucia Fuchslueger[1,2*], Emily F. Solly[3*], Alberto Canarini[1,4], Albert C. Brangarí[5,6]

[1]Center for Microbiology and Environmental Systems Science, University of Vienna, Djerassiplatz 1, 1030 Vienna, Austria
[2]Environment and Climate Hub, University of Vienna, Augasse 2-6, 1090 Vienna, Austria
[3]Helmholtz Centre for Environmental Research – UFZ, Permoserstr. 15, 04318 Leipzig, Germany
[4]Center for Ecological Research, Kyoto University, Kyoto, Japan
[5]Department of Ecosystem and Landscape Dynamics, Institute for Biodiversity & Ecosystem Dynamics, University of Amsterdam, Amsterdam, The Netherlands
[6]Institute for Physical Geography and Ecosystem Science, Lund University, Lund, Sweden. University of Lund, Lund, Sweden

*_Correspondence to_: Lucia Fuchslueger (lucia.fuchslueger@univie.ac.at) and Emily Solly (emily.solly@ufz.de)

**Abstract.** 'Global change' significantly alters organic matter and element cycling, but many of the underlying processes and consequences remain poorly understood. The interface of plants and soil plays a central role, coupling atmosphere, biosphere
and lithosphere and integrating biological and geochemical processes. The contributions to this special issue tackled questions on both biotic and abiotic interactions underlying responses of terrestrial biogeochemical cycling to a range of global changes, including increases in atmospheric $CO_2$ concentrations, warming, drought and altered water regimes. In this this overview we provide insights into the empirical, conceptual, and modelling-based studies featured in this special issue. In the following, we synthesize key findings covering 1) responses of plants to elevated $CO_2$, 2) the role of soil organisms in
modulating responses to warming, 3) impacts of global change on soil organic carbon, nitrogen, and mineral nutrient availability and 4) the influence of altered water table-depth caused by global change on greenhouse gas emissions (Figure 1). We showcase studies conducted in regions from the arctic to the tropics and highlight the manifold impacts of global change on various ecosystem components controlling biogeochemical processes occurring at the plant-soil interface. This understanding is crucial for deciphering feedbacks of terrestrial ecosystems to the climate system.



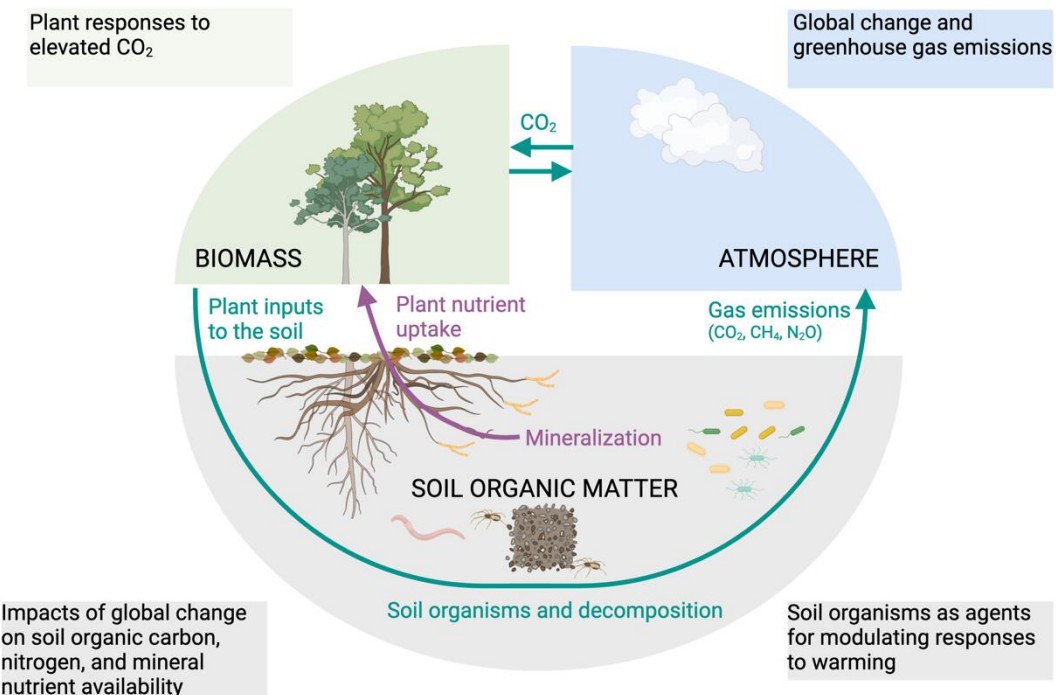


**Figure 1:** Global change can trigger changes across all ecosystem components. Altered environmental conditions have the potential to influence plant productivity and C allocation to maintain resource and nutrient acquisition, which can change both quality and quantity of C inputs to the soil. Similarly, global change can modify community structure and activity rates of soil organisms, and subsequently soil organic matter decomposition and weathering rates, overall affecting soil nutrient

availability. Finally, greenhouse gas emissions to the atmosphere respond sensitively to global change, thereby feeding back on the climate system. The figure was created with BioRender.com.

**Introduction**

Global change poses a major threat creating multiple environmental challenges significantly impacting biogeochemical cycling. Increased atmospheric carbon dioxide ($CO_2$) concentrations, primarily driven by human activity, have strongly

altered the Earth's energy budget, resulting in unprecedented global warming and shifts in precipitation patterns. The last decade has witnessed the hottest years on record, accompanied by intensified droughts and an increased frequency of extreme weather events (Intergovernmental Panel on Climate Change, 2023). These strong environmental changes significantly impact terrestrial ecosystems through a complex interplay of direct and indirect effects (Bardgett et al., 2008). Elevated $CO_2$ and temperatures can directly stimulate plant growth, but also trigger a series of indirect effects, so that



subsequent impacts on soil organic carbon (SOC) sequestration and total ecosystem carbon (C) budgets remain unclear. Likewise, changing moisture and temperature can directly affect soil microorganism activity and SOC decomposition (Kim et al., 2012; Moyano et al., 2013; Verbrigghe et al., 2022), but also indirectly influence soil nutrient availability, SOC stocks and plant feedbacks (Melillo et al., 2011). The magnitude and direction of responses to different global change factors depends largely on the ability of living organisms to tolerate or adapt to changed conditions (Allison, 2023). As greenhouse

gas emissions to the atmosphere are driven by autotrophic and heterotrophic organisms, understanding the resistance and resilience of all components in an ecosystem to changing conditions is integral for assessing potential feedbacks on the climate system. Overall, the plant-soil interface stands out as a hotspot for assessing these feedbacks, as it couples atmosphere, biosphere and lithosphere and integrates biological and geochemical processes across multiple trophic levels. Modelling approaches can help understanding current conditions, and predicting future ecosystem feedbacks to

environmental and climate change, particularly through altered biogeochemical cycling (e.g. Medlyn et al., 2015). However, the efficacy of models relies on a robust mechanistic understanding of the underlying physiological and biogeochemical processes.

The contributions to this special issue 'Global change effects on terrestrial biogeochemistry at the plant-soil interface'

covered topics ranging from 1) the ability of plants to regulate resource optimization in response to elevated atmospheric $CO_2$, 2) highlighting the role of soil organisms as agents for modulating responses to warming with a focus on artic ecosystems, 3) the effects of climate change on soil organic C, nitrogen (N) and mineral nutrient availability, and 4) the impact of changing soil water-tables on greenhouse gas emissions from boreal to tropical ecosystems (Figure 1). Importantly, the contributing studies couple knowledge originated from understudied ecosystems, encompassing studies

conducted in arctic, high-latitude, dry- and wetland ecosystems, which are not only potentially largest soil C sinks and can sensitively respond to global change, and provide new insights and more comprehensive understanding of the susceptibility of Earth's ecosystems to global change.

## 1 Plant responses to elevated $CO_2$

Plants are a major component in terrestrial C cycling, taking up $CO_2$ from the atmosphere and converting it into biomass.

However, the extent and mechanisms of plant resistance and recovery to global change remain uncertain. Exposure to changing environmental conditions, such as increased $CO_2$ concentrations, warming or drought episodes, can induce both short (e.g., physiological) and longer-term (e.g., vegetation composition change) responses, with implications for biogeochemical cycling at the ecosystem level. Especially, trade-offs between plant resource acquisition (water-use and nutrient uptake) and biomass growth can yield a substantial variation in climate change feedback. Using different modelling

approaches, Manzoni et al., (2022) explored whether elevated atmospheric $CO_2$ concentrations could lead to reduced transpiration rates from vegetation, potentially reducing water loss and facilitating ecosystem 'greening' in dry regions. They





found, however, small variation in canopy-level transpiration with rising atmospheric $CO_2$ due to the close coordination of physiological (reduced stomatal conductance) and morphological (increased leaf area) characteristics of plants to maximize water use efficiency. Yet, the responses to elevated atmospheric $CO_2$ are also tightly modulated by nutrient resource

availability (Walker et al., 2021). In a field Free-Air-$CO_2$-enrichment experiment in a mature Eucalyptus forest, Pihlblad et al., (2023) observed a faster nutrient recycling, particularly of available phosphorus forms, but not an additional availability of nutrients through soil organic matter (SOM) decomposition. In the long term, this could lead to plant nutrient limitations and reduce the potential of mature forest ecosystems to fix C as biomass in response to elevated $CO_2$. Both studies (Manzoni et al., 2022; Pihlblad et al., 2023) highlight the importance of accounting for resource limitations and respective adaption by

plants to accurately predict ecosystem responses to climate change.

## 2 Soil organisms as agents for modulating responses to warming

Global warming is one of the most pronounced climate change factors driving large ecosystem transformations. This is particularly evident in cold ecosystems, with the Arctic containing 50-70% of the total C stock (Hugelius et al., 2014) being crucial for terrestrial C cycling. Predictions suggest that Arctic regions will experience greater temperature increases

compared to other ecosystems, underscoring the importance to understand the unique challenges faced by these regions (Intergovernmental Panel on Climate Change, 2023). Soil (micro-)organisms are key regulators of matter and energy cycling between the soil and the atmosphere via their catabolic and anabolic processes (Ågren and Wetterstedt, 2007). Given the temperature sensitivity of enzymatic reactions, warming has mostly a stimulating effect on microbial activity in cold ecosystems, which can potentially destabilize C stored in Arctic soils and contribute to further climate change. Thus,

investigating warming effects on soil microbes in cold ecosystems is particularly relevant. Rijkers et al., (2023) identified strong differences in the temperature adaptation of microbial communities in soils from the subarctic to the High Arctic region. They predicted that summer warming can increase the community-level optimal growth temperature, but found no evidence supporting temperature adaptation of the soil bacterial community structure. SOM decomposition is not only strongly affected by microbial communities, but also by the presence of soil fauna. Blume-Werry et al., (2023) proposed that

in Arctic ecosystems climate warming can create new niches for soil organisms migrating both laterally and vertically belowground. This may introduce new soil-fauna-associated functions that can strongly affect decomposition processes and challenges the way we simulate C losses under climate change scenarios. Indeed, Monteux et al., (2022) showed that introducing a model microarthropod (Collembola *Folsomia candida*) species can influence bacterial diversity, and prime respiration of the native soil community. These studies highlight the importance of investigating and understanding biotic

interactions in soil and to further explore potential additive or interactive effects of warming (and other climate change factors) on the soil food web.



## 3 Impacts of global change on soil organic carbon, nitrogen, and mineral nutrient availability

Warming is expected to boost net primary production and potentially enhance ecosystem C sequestration (Melillo et al., 2017). However, warming also has a stimulating effect on microbial activity and SOM turnover (e.g., Classen et al., 2015;

Davidson and Janssens, 2006), which can lead to significant decreases in soil organic C stocks (Verbrigghe et al., 2022). Soil microbes release enzymes transforming SOM into available C and nutrients, ultimately releasing $CO_2$ via intracellular metabolism (Allison et al., 2007). Many studies have examined the temperature sensitivity of SOM decomposition via $CO_2$ emission or mass loss of C substrate (e.g., Kirwan et al., 2014; Wang et al., 2013). However, differentiating contributions by extracellular and intracellular processes is key for a mechanistic understanding of warming. Adekanmbi et al., (2023)

investigated these sensitivities and found that depolymerization of high molecular weight C was more sensitive to temperature changes at higher temperatures, whereas the respiration of the generated monomers was more sensitive to temperature changes at moderate temperatures. Hence, this study suggested that global warming may shift the importance from extracellular depolymerization to intracellular metabolic processes as the rate-limiting step of SOM mineralization.

The stimulating effect of warming on plant productivity can be constrained by low nutrient availability, for instance by N (Melillo et al., 2011) and cations (Jonard et al., 2015). In cold ecosystems, the addition of nutrients has been found to stimulate plant aboveground growth (Bergh et al., 2008), but to reduce the allocation of C to roots and associated microorganisms (Wallander et al., 2011). Almeida et al. (2022) investigated the impact of forest fertilization on the soil ectomycorrhizal fungal community in an experimental spruce forest in Norway. The authors observed changes in the fungal

community composition after three years of N additions, potentially affecting hydrophobic SOM formation and reducing soil C sequestration. Myers-Pigg et al. (2023) explored the net response of SOC composition to climate change, assessing its diagenetic state based on lignin-phenol biomarkers. Their biomarker-based approach revealed a coupled increase in C and N cycling rates, which may support forest productivity and help maintain SOC stocks. Finally, in a modelling study using the ForSAFE model on different forest systems distributed across Sweden, Kronnäs et al. (2023) found that warming and longer

growing seasons for plants could enhance soil weathering rates, inducing substantial changes in soil texture and mineral nutrient stocks. This emphasizes the importance of including soil texture and seasonal climate variations for prediction forest responses to climate change.

## 4 Influence of altered water-table depth caused by global change on greenhouse gas emissions

One of the most effective and accessible approaches for evaluating effects of global changes on soil biogeochemical cycling

is the measurement of gaseous emissions, such as of carbon dioxide ($CO_2$), methane ($CH_4$), and nitrous oxide ($N_2O$). From soils, $CO_2$ is released by plant roots and SOM decomposition by soil organisms under aerobic conditions (autotrophic and heterotrophic respiration), $CH_4$ under anaerobic conditions by methanogenic microorganisms, and $N_2O$ through nitrification



and denitrification processes in soils. All gas fluxes are highly responsive to changes in soil water conditions controlling whether ecosystems are sources or sinks for greenhouse gases (Jungkunst and Fiedler, 2007). Zhao and Zhuang (2023)

modelled greenhouse gas flux responses of Northern peatlands in response to warming, revealing increases in both $CO_2$ and $CH_4$ emissions due to deeper water-table levels and enhanced permafrost thawing. They demonstrated that warming had limited impact on net primary production due to shifts in plant functional types and N limitations, which could turn ecosystems into C sources. Their estimates proposed that this shift might occur as early as 2050, notably earlier than previously anticipated (e.g., Gallego-Sala et al., 2018; McGuire et al., 2018). Zhang et al., (2022) further explored the effect

of water-table levels and N availability on greenhouse gas emissions in alpine peatlands by conducting a mesocosm experiment. They found that a moderate increase in N deposition and rising water-table levels stimulated $CH_4$ emissions, whereas only N deposition consistently increased $N_2O$ emissions. Finally, in a field study, Castellón et al., (2022) quantified the emissions of $CO_2$ and $CH_4$ in a natural mangrove forest experiencing daily and seasonal variations in water-table depths. Higher topography areas generally exhibited increased $CO_2$ emissions and decreased $CH_4$ emissions, likely induced by

reduced flooding frequency. The study highlighted the role of microtopography, seasonality and soil physicochemical properties in influencing greenhouse gas emissions.

**Concluding remarks**

The studies contributing to this special issue 'Global change effects on terrestrial biogeochemistry at the plant-soil interface', reveal pronounced impacts of global change on biogeochemical processes driven by different trophic levels, extending from

autotrophic plants to heterotrophic soil fauna and microbial populations. The related effects are particularly relevant in understudied regions such as the arctic and the tropics. Both empirical and modeling studies show that global change can affect small soil-scale, as well as large ecosystem-scale fluxes of C and nutrients, inducing substantial feedback loops on the climate system. The richness of methodologies employed in these studies highlights that a combination of techniques is essential for mechanistically and holistically understanding the complex phenomena occurring at the plant-soil interface, and is crucial for

improving predictions of the resistance and resilience of terrestrial ecosystems to global change.

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



## Acknowledgements


We want to thank Michael Bahn for initiating and supporting this special issue. LF acknowledges the European Union's Horizon 2020 research and innovation program under the Marie Sklodovska-Curie grant agreement No. 847693 (REWIRE). EFS acknowledges funding from the Swiss National Science Foundation (Ambizione; grant no. PZ00P2_180030). AC received financial support as an International Research Fellow of JSPS [Postdoctoral Fellowships for Research in Japan

(Standard)]. AB received support from the Swedish Research Council Formas (grant no. 2022-01478).

**Competing interests:** At least one of the (co-)authors is a member of the editorial board of Biogeosciences