# Peer review of "Overview: 'Global change effects on terrestrial biogeochemistry at the plant-soil interface'"

_EGUsphere, 2023_

## Author Response (AR1)

**Comments to the referees:**

**Reviewer 1:**
This is an editorial for a special issue "Global change effects on terrestrial biogeochemistry at the plant-soil interface" at Biogeosciences. I find the topic is interesting and well written, and there are only some minor places which need to revised before publication.

We thank the referee for the positive evaluation and for the helpful comments, which we have addressed in our revised version.

Line 33 Rephrase this sentence
Has been changed to:
Line 27: *'Global change poses manifold threats on different components of ecosystems and can significantly impact biogeochemical cycling.'*

Line 61 and change to but also
Has been changed as suggested.

Line 100 so as to further explore…
This has been changed to:
Line 109: *'…interactions in soil as well as the need to further explore potential additive or interactive of warming and other climate change factors on the soil food web.'*

Line 154 mechanistic and holistic..
Has been changed as suggested.

**Reviewer 2**
In this manuscript, the authors present a comprehensive overview of publications in the special issue "Global change effects on terrestrial biogeochemistry at the plant-soil interface", revealing how global changes such as increased atmospheric CO2, warming, drought, and altered water regimes impact terrestrial biogeochemical cycling at the plant-soil interface. Overall, the manuscript was well-written and organized.

We thank the referee for the positive evaluation and for taking time to provide some comments. We have addressed them accordingly in our revised version.

Here are some minor comments:
L18: delete "this" and add a comma after "overview"
This was changed as suggested

I would suggest moving Figure 1 to the context rather than before the INTRODUCTION.
We have moved Figure 1 after the introduction.

L95: add a comma after "proposed that", "ecosystems", and "organisms"
This was changed as suggested.

L97: challenge
This was not changed as challenges referes to 'This may introduce..' at the begin of the sentence.

L126: predicting
This was changed as suggested.

Consider including a section on future research directions or unanswered questions to guide subsequent studies in this rapidly evolving field.
We thank the reviewer for this comment, and agree that providing some guidelines for future directions is helpful in this field of research. We have now pointed to these at the end of each main section of the overview.

**Reviewer 3:**
This overview comprehensively and systematically summarizes global change effects on terrestrial biogeochemistry at the plant-soil interface based on topics in the special issue, presenting its content in a clear manner. It addresses a highly significant ecological and environmental issue worthy of attention. Based on the content of the overview, I have only a few minor questions and suggestions which need to be further considered.
We are grateful for the careful evaluation of our special issue overview.

**Plant responses to elevated CO2:** It may be crucial to add some perspectives regarding different vegetation types in this section or concluding remarks. Different plant species possess varying physiological characteristics and response mechanisms, which influence their sensitivity and adaptability to climate change. This discrepancy is particularly notable when considering significant vegetation types such as crops and forests. For crops, they are typically subject to direct anthropogenic influences, including planting methods, fertilization, and irrigation. These factors interact with climate change, further impacting their growth and yield. Therefore, in addition to considering the effects of climate change, anthropogenic management factors also need to be taken into account for crops. On the other hand, forests, as a crucial vegetation type, exhibit distinct regional distribution characteristics. Different types of forests respond differently to climate change; for example, temperate forests and tropical rainforests possess different ecosystem structures and functionalities. Hence, targeted research is necessary to understand their response mechanisms to climate change. Therefore, the author may consider delving into a more in-depth discussion and comparison of the responses of different vegetation types to provide a more comprehensive understanding of the impact of climate change on the soil-plant interface.
We fully agree with the reviewer that responses to changing climate conditions depend strongly on species, plant type and vegetation type (managed or naturally growing) and may not be totally generalizable but is more nuanced. However, we on purpose did not want to extend this overview into a literature analysis but provide an integrated view of the studies that contributed to this article collection, therefore we also tried to narrow down our interpretations. However, we agree that it is important that those points need to be considered and added the following sentences:

Line 70: *'Plants are a major component in terrestrial carbon cycling, taking up CO2 from the atmosphere and converting it into biomass. However, the extent and mechanisms of plant resistance and recovery to global change depends on many biotic and abiotic factors. For instance, responses can strongly differ between vegetation types (e.g. crop plants vs. forest stands), physiological traits (e.g., wet vs. drought adapted species) and plant ecological strategies and flexibility (e.g. specialists vs. generalists), as well as depend on the magnitude of the environmental change.'*

Line 85: *'Both studies (Manzoni et al., 2022; Pihlblad et al., 2023) highlight the importance of accounting for resource limitations of plants in response to elevated CO2, however also emphasize that adaptions can profoundly differ between ecosystems or plant communities. It is thus essential to improve our understanding of the variability in short- and long-term plant responses to accurately predict ecosystem behavior under climate change.'*

**Line 100:** The authors are suggested to compile these research topics with more discussions on the response of rhizosphere processes to climate change, which is a highly significant process involving interactions among soil, plants, and microorganisms. For instance, climate change may affect the distribution and availability of water in the soil, thereby influencing the growth of plant roots and rhizosphere processes. Extreme climate events such as droughts and increased precipitation may alter the water content and distribution in the soil, subsequently impacting root growth and rhizosphere processes. Additionally, climate change may also influence the temperature and nutrient cycling in the soil. Temperature fluctuations may affect the activity and metabolic processes of soil microorganisms, thereby influencing the decomposition of organic matter and nutrient cycling in rhizosphere processes. Furthermore, climate change may alter the nutrient content and availability in the soil, thereby affecting the ability of plant roots to absorb and utilize nutrients. Climate change may also impact the biodiversity and functionality of plants and microorganisms in the soil. The microbial communities and functions in rhizosphere processes may be affected by climate change, thereby influencing the efficiency and effectiveness of rhizosphere processes. Moreover, climate change may affect the growth and physiological processes of plants, thereby influencing the production of root exudates by plants and rhizosphere processes in the soil.

We highly appreciate this comment, however, as mentioned above the main scope of this Overview was to summarized global change effects on terrestrial biogeochemistry at the plant-soil interface based on the topics of the papers published in the special issue, rather than to provide an extensive literature review.

Specific comments:
Line 14: add "hydrosphere" as this overview mentions "transpiration" in Line 71
This was added as suggested.

Line 19: delete one of "this"
Thanks for spotting this typo, one 'this' was deleted.

Line 39: which kinds of "indirect effects"?
We here had in mind that elevated $CO_2$ does not necessarily change soil microbial decomposers, as they will maybe not directly respond to increased $CO_2$, but rather respond indirectly for instance to changed plant C inputs or other plant-mediated changes.

We have added here (now line 34): *'(e.g. changed carbon supply for soil microbial communities)'*

Line 54-62: This part largely repeated the content in the abstract, which may need revision.
We agree here, and modified both the abstract and last paragraph:

Abstract:

Line 19-23: *'In this overview, we synthesize key findings of the contributing empirical, conceptual, and modelling-based studies covering responses of plants to elevated $CO_2$, the role of soil organisms in modulating responses to warming, impacts of global change on soil organic carbon, nitrogen, and mineral nutrient availability and the influence of altered water table-depth caused by global change on greenhouse gas emissions.'*

Introduction:

Line 49-53: *'The contributions to this special issue 'Global change effects on terrestrial biogeochemistry at the plant-soil interface' could be categorized in four functional topics. Studies investigated 1) the ability of plants to regulate resource optimization in response to elevated atmospheric $CO_2$, 2) the role of soil organisms as agents for modulating responses to warming, with a strong focus in particular on artic ecosystems, 3) the effects of climate change on soil organic carbon, nitrogen and mineral nutrient availability, and 4) the impact of changing soil water-tables on greenhouse gas emissions from boreal to tropical ecosystems (Figure 1).'*

Line 76: delete the comma in "Manzoni et al., (2022)","Pihlblad et al., (2023)" etc.
Thanks for pointing this out, we have changed this accordingly.

Line 78: "to fix C"----->"to fix carbon", also replace "C" "N" with "carbon" "nitrogen" in other places
This was changed throughout the manuscript as suggested.

Use the same tense for "highlight", "suggested", "emphasizes", "highlighted" in the summary sentences
This was double checked and changed if necessary.